# Effect of Gellan Gum/Tuna Skin Film in Guided Bone Regeneration in Artificial Bone Defect in Rabbit Calvaria

**DOI:** 10.3390/ma13061318

**Published:** 2020-03-14

**Authors:** Seunggon Jung, Hee-Kyun Oh, Myung-Sun Kim, Ki-Young Lee, Hongju Park, Min-Suk Kook

**Affiliations:** 1Department of Oral and Maxillofacial Surgery, School of Dentistry, Chonnam National University, Gwangju 61186, Korea; seunggon.jung@jnu.ac.kr (S.J.); hkoh@jnu.ac.kr (H.-K.O.); omspark@jnu.ac.kr (H.P.); 2Department of Orthopaedics, School of Medicine, Chonnam National University, Gwangju 61469, Korea; mskim@jnu.ac.kr; 3School of Chemical Engineering and Biocosmos Co., Chonnam National University, Gwangju 61186, Korea; kilee@jnu.ac.kr

**Keywords:** gellan gum, tuna skin gelatin, guided bone regeneration, membrane

## Abstract

It is necessary to prevent the invasion of soft tissue into bone defects for successful outcomes in guided bone regeneration (GBR). For this reason, many materials are used as protective barriers to bone defects. In this study, a gellan gum/tuna skin gelatin (GEL/TSG) film was prepared, and its effectiveness in bone regeneration was evaluated. The film exhibited average cell viability in vitro. Experimental bone defects were prepared in rabbit calvaria, and a bone graft procedure with beta-tricalcium phosphate was done. The film was used as a membrane of GBR and compared with results using a commercial collagen membrane. Grafted material did not show dispersion outside of bone defects and the film did not collapse into the bone defect. New bone formation was comparable to that using the collagen membrane. These results suggest that the GEL/TSG film could be used as a membrane for GBR.

## 1. Introduction

Bone regeneration is necessary after bone loss caused by tumor ablation, congenital defects, fractures, and other issues [1,2] in the field of oral and maxillofacial surgery, as well as in many other medical fields. The size of the bone defect, soft tissue coverage of the defect, and surgical method significantly determine the success of bone regeneration [3]. In bone regeneration, the bone defects need to be isolated from the surrounding soft tissues to prevent unwanted connective tissue from growing into the defective bone. The materials for isolation are usually membranes made of various materials; they could be degradable or non-degradable. It is necessary to remove the latter after bone regeneration. In guided bone regeneration (GBR), a membrane is used to prevent soft tissue invasion, which can hinder bone regeneration [4]. GBR procedures are commonly performed to repair bone defects arising from pathologic lesions or augment alveolar bone before dental implant surgery [5]. The role of the membrane is crucial in GBR. It prevents the soft tissue from invading the bone defect and preserves the space of the bone defect during bone regeneration. The GBR membrane should have the following characteristics: (1) biocompatibility; (2) proper stiffness for space maintenance; (3) ability to prevent epithelial cell migration; and (4) appropriate resorption time for proper bone regeneration [4]. The success of GBR depends on the resorption rate of the membrane and its time-effectiveness as a barrier [6].

Gellan gum (GEL), a bacterial polysaccharide, is widely studied and described. It is produced by the bacteria Sphingomonas elodea [7]. GEL has been investigated for application as a food ingredient or additive for pharmaceutical products [8]. Since GEL has good heat resistance and enzyme resistance, it is also evaluated for tissue engineering applications [9]. There are many studies of GEL for tissue engineering, such as wound dressing, artificial cartilage, and osteogenesis [10,11,12]. GEL can be used in various forms, including film, hydrogel, microcapsules, and sponge, and its constructs can also be applied in GBR [13]. GEL can also be applied to improve poor mechanical properties of other materials applied in bone reconstruction [14]. However, GEL, an anionic polysaccharide, can inhibit cell attachment [15]. By blending GEL with bio-informative materials, regular cell behavior can be induced [16,17].

Gelatin derived from degradation of natural collagen has wide applications in the food and pharmaceutical industries and is mainly acquired from porcine and bovine sources. However, such materials may not be accepted by Muslim and Jewish people for religious reasons. In such circumstances, fish gelatin and/or collagen materials could be an alternative [18,19]. It is reported that gelatin from tuna showed higher gel strength and similar viscoelastic properties compared to mammalian gelatins [20]. However, gelatin in membrane form is hard to use as a GBR membrane due to its mechanical weakness.

GEL and gelatin have been studied to build a scaffold for bone regeneration or 3D culture of cardiomyocytes. [14,17]. However, GEL products of complicated modification could be expensive for commercial use; more simple forms of GEL might be more appropriate to meet clinical demand. In this study, we prepared a mixture of GEL and gelatin from tuna skin and fabricated it in the form of a film to use as a barrier for GBR. In contrast to using GBR as a scaffold, GBR membranes do not need to exhibit a high level of cell attachment. To investigate the feasibility of use of the film as a GBR membrane, we evaluated its properties and effects on bone regeneration in animal experiments.

## 2. Materials and Methods

### 2.1. Preparation of Tuna Skin Gelatin (TSG)

Tuna (Thunnus albacares) skin was trimmed and soaked in 0.2 N acetic acid for 6 h at a low temperature (4~8 °C) and neutralized in running water for 12 h. After adding water four times to the neutralized tuna skin, the skin was heated in hot water at 60–70 °C for 3–4 h. Impurities were removed through filtration. The filtered solution was heated at 60–70 °C and reduced to the original weight of the trimmed tuna skin. Then, dialysis was performed using distilled water (1 L, 2 times, ~10 h/time) at 4 °C using a membrane tube with a cutoff size ranging from 12 to 14 kDa. To obtain a powdered form, freeze-drying was performed to eliminate most impurities except protein. The tuna skin gelatin (TSG) extract powder was stored in a desiccator.

### 2.2. Preparation of Gellan Gum and GEL/TSG

The gellan gum and TSG (GEL/TSG) forming solutions were prepared by dissolving the TSG and gellan gum granules (GEL; Gelzan™, Mw = 1000 kg/mol, CP Kelco, Atlanta, GA, USA) in distilled water at 90 °C and 200 rpm. The solution was cast onto a Petri dish and dried at 50 °C for 24 h. After applying phosphate-buffered saline for 6 h, the film was washed with distilled water.

### 2.3. MTT Assay of GEL/TSG

The cell survival rate measurement was determined using the MTT assay method. The GEL and GEL/TSG were prepared for the MTT assay, following which they were added to the cell culture fluid (DMEM, Thermo Fisher Scientific, Waltham, MA, USA) and dissolved for more than 24 h at 37 °C to acquire an eluent solution. NIH3T3 fibroblasts were seeded on a 96-well plate at a density of 1 × 104 cells per well and incubated for 24 h. The cells were treated using prepared eluent solutions of various concentrations for 24 h. The culture medium in each well was replaced with 20 μL of 5 mg/mL stock solution of 3-(4,5-dimetylthiazol-2-yl)-2,5-diphenyltetrazolium bromide (MTT; Sigma-Aldrich, St. Louis, MO, USA). The cells were incubated for 4 h at 37 °C in a humidified incubator under a 5% CO_2_ atmosphere. After the supernatants were removed, the formazan crystals were dissolved in 100 μL of DMSO. Using the ELISA machine, the optical density of the film was measured at 570 nm (ELX 808, Biotek Instruments, Winooski, VT, USA).

### 2.4. Animal Experiment

To evaluate the effectiveness of the GEL/TSG film as a membrane, we performed GBR on bone defects artificially prepared on the parietal bones of rabbits. The graft material used was beta-tricalcium phosphate (KJ Meditech, Gwangju, Republic of Korea); a collagen membrane (Lyoplant, B. Braun, Melsungen, Germany) was used as the positive control group. The experimental design was as follows. Negative control group: (A) bone graft (BTCP) without a membrane; (B) positive control group: bone graft with the collagen membrane; (C) experimental group: bone graft with the GEL/TSG film as the membrane. This study was performed in accordance with the prescribed guidelines of the Chonnam National University Institutional Animal Care and Use Committee (CNU IACUC-H-2014-7). Sixteen domestic rabbits were anesthetized with 10 mg/kg of Xylazine (Rompun, Bayer Korea, Seoul, Korea) and 50 mg/kg of Zoletil (Zoletil50, Virbac, Carros, France). The hair on the rabbits’ parietal scalp was shaved and disinfected using a povidone-iodine solution (Potadine, Samil, Seoul, Korea); injection of 2% lidocaine with 1:100,000 epinephrine was done on the rabbits’ parietal scalp. Incision was made along the mid-sagittal plane using a No. 15 blade to expose the parietal bone of the rabbits, following which bone defects were created by performing ostectomies on both parietal bones using a 10 mm trephine bur. BTCP graft material was inserted into the bone defects and secured with the membrane. The surgical sites were sutured with 3–0 Vicryl (Vicryl, Ethicon, Livingston, UK). Postoperatively, a prophylactic antibiotic (Fortimicin, Young Jin Pharm, Seoul, Korea) and a non-steroidal anti-inflammatory drug (Fenaca, Hana Pharm, Seoul, Korea) were injected once daily for five days to prevent infection and to achieve analgesia.

### 2.5. Micro-Computed Tomography (Micro-CT) Analysis

At two weeks post-surgery, the rabbits were sacrificed by injecting them with excess pentothal sodium; the parietal bones were resected, including the bone graft site, using a bone cutter, and trimmed into a proper shape and size. The formation process of the new bone was observed using a radiographic apparatus (Hi-Tex, Osaka, Japan) at 35 kV and 400 mA (2D). The voltage and current of the X-ray source were set at 50 kV and 200 A, with a pixel size of 17.09 mm. The exposure time was 1.2 s. Over an angular range of 180 degrees (angular step of 0.4 degrees), four hundred and fifty projections were acquired. The image slices were reconstructed using 3D CT analyzer software (CTAn ver. 1.1; Skyscan, Kontich, Belgium).

### 2.6. Histological Evaluation of Samples

The rabbits were sacrificed, and the specimens were resected as described as above, at two and four weeks after the operational experiment. The specimens were soaked in formalin for two days and seeded in an EDTA solution. Following paraffin embedding, 5 µm tissue sections were prepared and dyed with hematoxylin and eosin before they were observed via optical microscopy (Nikon, Melville, NY, USA). Using AperioImageScope v9.1 (ImageScope, Aperio Technologies, Vist, CA, USA), we obtained histological digital images of the slides.

## 3. Results

### 3.1. MTT Assay

The cytotoxicity was evaluated by ISO 10993 protocol. The eluent solution, regardless of the concentration, manifested cell viability of more than 80% notwithstanding the presence of TSG, indicating that GEL and GEL/TSG have very low cytotoxicity (Figure 1).

### 3.2. Micro-CT Analysis

The grafted material was implanted in the bone defect in both the positive control group and experiment group. However, the grafted material was dispersed out of the bone defect in the negative control group, in which no membrane was implanted, even though the grafting procedure was the same as that in the experiment (Figure 2).

### 3.3. Histologic Assessment

Both the control collagen membrane and GEL/TSG film protected the bone defects from soft tissue invasion. Both the positive control group and GEL/TSG group exhibited new bone formation. In contrast, connective tissue grew in the negative control group which did not contain any membrane. In the specimen harvested in the second week, the graft material remained in place in the bone defects, and soft tissue growth was minimal in the positive control group. In the fourth week, it was observed that the collagen membrane of the positive control group was degraded. In the second week, it was observed that the GEL/TSG film was adjacent to the bone defect margin, as a result of which soft tissue invasion of the bone defect was prevented. New bone formation and the degradation of the GEL/TSG film was observed in the experiment group in the fourth week (Figure 3).

## 4. Discussion

Various experiments were conducted for the application of GEL in alveolar bone regeneration. The in vitro properties of GEL as a filling material for dental extraction sockets were evaluated [21]. GEL demonstrated higher in vitro stability and manifested superior blood absorption rate compared to the commercially available fillings; it also inhibited fibroblast migration. Wang et al. prepared GEL microspheres grafted with gelatin, designed to deliver living cells to the damaged tissue. They suggested that the gelatin-graft-GEL microcarriers can be beneficial to clinical regenerative medicine in musculoskeletal or dermatological fields [22]. Chang et al. fabricated GEL films of 1%, 1.5%, and 2% and applied 2% film to the animal GBR study model [13]. The bone defects in rats were covered with the film for two months. The GEL film prevented the soft tissue from penetrating the bone defects; it also exhibited the desired biodegradability without signs of inflammation in the surrounding tissues. Similarly, it was observed that partial degradation of the film was observed in second-week specimens and the degradation was more evident in fourth week specimens without sign of inflammation in histological examinations of our study.

Gelatin is a protein compound obtained from the hydrolysis of collagen; it is used in the pharmaceutical and medical fields because of the enzymatic biodegradability and biocompatibility it manifests in physiologic environments [23]. There are many studies on the utilization of gelatin obtained from fish bone or skin, a by-product of fish processing [24]. TSG is biocompatible and can be applied to wounds without chemical modification because it has adhesive properties and no cytotoxicity. It contains more than 60 ㎎/g of amino acid, 5 ㎎/㎖ of protein, and 20% of chondroitin sulfate [25]. Because chondroitin sulfate, in addition to being the most important component of extracellular matrix, is known to prevent aging and encourage tissue regeneration, it is expected to have good effects on tissue regeneration. It is observed that introduction of TSG to GEL did not lower cell viability of GEL in the results of MTT assays in this study.

Titanium mesh is used in large bone defects. As titanium is metal, it makes for superior space maintenance. However, it is not popular in small alveolar bone defects because of the difficulty in manipulating and removing it. Degradable membranes are more widely used in smaller alveolar defects. Degradable membranes should be hydrolyzed and absorbed in the body without inflammatory reaction, and macrophages from foreign body reactions should not be observed during the degradation process. Bovine collagen membranes in GBR have demonstrated a soft tissue exclusion effect [26]. Nevertheless, bovine membranes do not maintain the required space when there is no graft material in the bone defect beneath the membranes because of lack of rigidity [27,28]. In a rat model study, it was stated that calvarial critical size defects covered by collagen membranes healed completely at four months without graft materials, although in most of them, there were repaired bones with concave appearance, indicating less vertical gain due to membrane collapse [29]. However, many of the degradable membranes in the market have low physical strength and a rapid degradation rate, which prevents foreign body reaction [30]. There have been many studies on enhancing the mechanical properties and degradation rates by various methods of inducing cross-linking-agents or mixture of polymers [28]. Barbani et al. reported that gellan strengthened the mechanical properties of complex nanocomposite scaffolds containing hydroxyapatite and gelatin [14]. There were also studies of fillers for mechanical improvement of GEL. Graphene oxide was combined with GEL and made into composite films which showed combined interactions of coordination bonding, ionic bonding, and hydrogen bonding and thus provided good mechanical performance [31].

In this study, we attempted to simplify GBR by creating a composite of GEL and TSG; we assessed its effectiveness for preventing soft tissue penetration into the bone defect, which hinders bone regeneration. We evaluated the design on a rabbit calvarial model. Many mammalian cells need to be attached to a matrix material, and it is better to facilitate such an attachment for the cells to promote proliferation. However, because GEL is a relatively bio-inert material, anchorage-dependent cells grown in pure GEL matrices may exhibit low levels of cell attachment. Therefore, it is necessary to modify GEL such that it can be used as an artificial extracellular matrix [16]. GEL can be blended with bio-informative materials that induce regular cell behavior. Koivisoto et al. reported that functionalized GEL with gelatin could be the material for biomimicking scaffolds for 3D culture of human cardiomyocytes [17]. Because GEL does not provide attachment for cells, they chose the biofunctionalization of GEL with introduction of gelatin. Cencetti et al. described the preparation of silver loaded wound dressing based on GEL and hyaluronic acid, which showed enhanced water uptake capability and slow dehydration rates [10]. They also reported that a hydrogel of GEL and hyaluronic acid could be used for prevention of epidural fibrosis formation [32]. Cerqueira et al. proposed loading of human adipose stem cells and microvascular endothelial cells on GEL and hyaluronic acid hydrogels and reported the hydrogels promoted neovascularization [33]. Lactoferrin was also used for bone tissue engineering. Bastos et al. integrated bioactive factors of lactoferrin and hydroxyapatite to GEL sponge-like hydrogels and observed sustained release of lactoferrin and increased human adipose stem cell viability [34].

Gelatin has also been integrated with GEL to improve the latter’s mechanical strength and biocompatibility [35,36]. It is reported that blending of GEL and gelatin resulted in a synergistic increase of gel network strength and gel firmness [37,38]. GEL has a hard and brittle form, while gelatin forms a soft, flexible, and elastic form [39]. Further, GEL and gelatin showed synergism at particular ratios and salt concentrations [40]. It is also important to improve GEL’s mechanical integrity, because hydrogel with insufficient inherent strength is not very effective in tissue engineering [41]. Previous studies of gelatin from fish mainly focused on pharmaceutical or nutraceutical use [42,43]. These support that gelatin from fish is edible, but there is lack of study about surgical use of gelatin from fish which supports biocompatibility. Since the normative framework and guidelines for experiments like ISO 22803:2004(en) are necessary to check biocompatibility before application to humans, evaluating the biocompatibility of GEL/TSG would be the focus of our future study.

Many of these prior studies were to investigate the possibility of application of GEL or gelatin as a bone substitute and focused on enhancing the mechanical strength or bone formation as a scaffold. We performed this study to evaluate the properties of GEL/TSG as a membrane for GBR, which demands less mechanical strength. Our results showed GEL/TSG film seemed to have the properties required for GBR membrane use.

## 5. Conclusions

In this study, we fabricated a film using a mixture of GEL and TSG and evaluated its characteristics and applicability to GBR as a membrane by performing an animal experiment. MTT assays confirmed that all the films were non-toxic, and the survival rate of cells was 80% or more, even with the addition of TS. After four weeks, we took micro-CT scans of the specimens. The grafted bone in the group treated with the GEL/TSG membrane held in bone defects, and bone regeneration was observed. In histologic examinations, similar results confirmed that the film had a positive effect on the formation of new bone, and degradation of the film was observed. From the results, GEL/TSG film could be considered as a viable candidate for a membrane in GBR.

## Figures and Tables

**Figure 1 materials-13-01318-f001:**
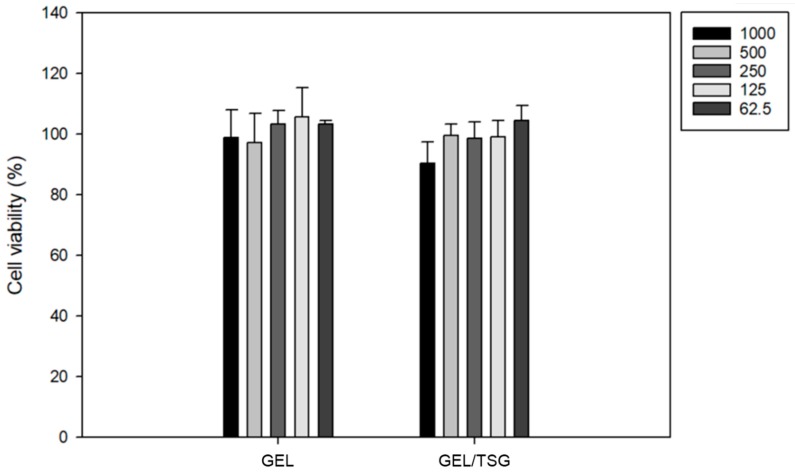
MTT assay of only gellan gum (GEL) and GEL/tuna skin gelatin (TSG). Cell viability remained unchanged.

**Figure 2 materials-13-01318-f002:**
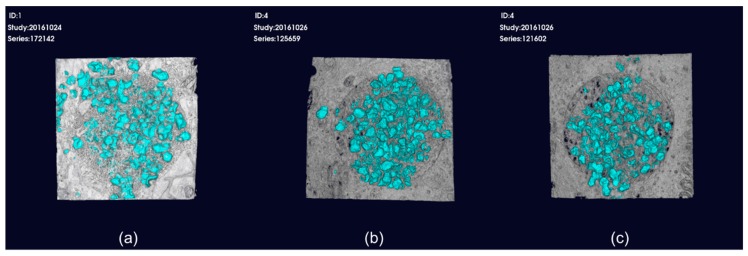
3D reconstructed micro-computed tomography (micro-CT) images. (**a**) Bone graft with no membrane. The grafted material is dispersed out of the bone defect; (**b**) bone graft with the collagen membrane; (**c**) bone graft with GEL/TSG film. Grafted material is implanted in the bone defect in (**b**) and (**c**).

**Figure 3 materials-13-01318-f003:**
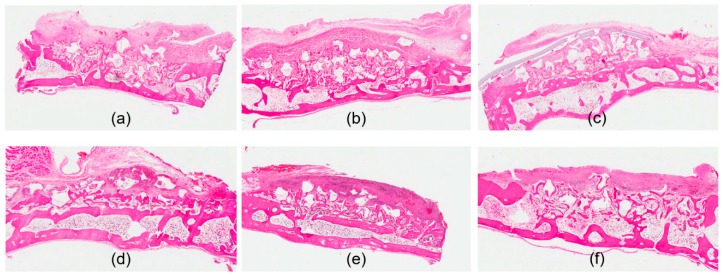
Histological images with hematoxylin and eosin stain. (**a**) Bone graft with no membrane at 2 weeks. New bone formation level is below the adjacent bone margin, and there is soft tissue invasion; (**b**) bone graft with the collagen membrane at 2 weeks. Elevated bone level is noted with the collagen membrane; (**c**) bone graft with the GEL/TSG film at 2 weeks. Presence of the GEL/TSG film and elevated bone level is observed; (**d**) bone graft with no membrane at 4 weeks. Thickened connective tissue over newly formed bone is observed; (**e**) bone graft with the collagen membrane at 4 weeks. Degeneration of the membrane and more advanced maturation of newly formed bone are observed; (**f**) bone graft with the GEL/TSG film at 4 weeks. Maturation of newly formed bone is observed. Degeneration of the film is noted.

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
