# Peer review of "Effect of Gellan Gum/Tuna Skin Film in Guided Bone Regeneration in Artificial Bone Defect in Rabbit Calvaria"

_materials, 2020, doi:10.3390/ma13061318_

Round 1

Reviewer 1 Report

Dearest Authors, you surely target an interesting research area, as bone ins the second most transplanted tissue after blood.

However IMHO your work needs substantial update prior to publication consideration:

1) it is not clear which is the potential clinical indication you target. is it dental GBR? orthopedy too or not? etc.

2) there are no explicit references to possible traslability into human of these materials, no comments of suitability of supplied raw materials for possible future human investigations etc (see for example EU-MDR requirements or USA-FDA ones).

3) there are no reference to normative framework and guidelines followed during experiments performed (e.g. ISO10993 etc)

4) discussion and conclusions are too concise and poor

5) literature is not updated, all new papers missing from almost all last 5 years (some key for example Haugen et all, J Periodont, 2019)

best

Author Response

1) it is not clear which is the potential clinical indication you target. is it dental GBR? orthopedy too or not? etc.
- The potential clinical indication is dental GBR. We clarified that in abstract and introduction.

2) there are no explicit references to possible traslability into human of these materials, no comments of suitability of supplied raw materials for possible future human investigations etc (see for example EU-MDR requirements or USA-FDA ones).
-Yes, there is no references that states raw material is suitable or acceptable for human use yet. We are investigating the applicability to bone regeneration of various materials. Yet the result of GEL/TSG was fair, its suitability to human is planned in future after comparing with other materials.

3) there are no reference to normative framework and guidelines followed during experiments performed (e.g. ISO10993 etc)
-We did not followed normative framework or guideline because this study is in experimental step, which is not yet for standardized manufacture.

4) discussion and conclusions are too concise and poor
-We augmentd discussion and modified conclusion.

5) literature is not updated, all new papers missing from almost all last 5 years (some key for example Haugen et all, J Periodont, 2019)
- Unfortunately, as the main materials, gellan gum and gelatin is not novel, there is not much of recent literatures. In our study, we expected novel use of these two old materials by making the mixutre. By the way, I appreciate your recommendation, but the literature you mentioned is about bone substitute.

Reviewer 2 Report

This work demonstrates a gellan gum/tuna skin gelatin scaffold for guided bone regeneration application. The tuna skin gelatin was chosen as a material of choice because of wider acceptance across community and religions. The composite film was cast via coacervation technique by mixing gellan gum and gelatin solution. The cytotoxicity analysis was performed to show the biocompatibility of the film. The 3D CT micro scan and histology show a better bone-forming ability of the gellan gum/tuna skin gelatin film.

Major Comment:

  1. The primary focus of this study seems to improve the mechanical strength of gelatin by adding gellan gum. However, there was no experiment to show that the proposed approach increased the mechanical strength of the film. It is recommended to include data that shows the mechanical properties of the film before and after modification.
  2. The proposed film needed to be characterised thoroughly. In the manuscript, however, there was no data or explanation related to characterisation. It would be good to include the degradation studies in the revised manuscript.
  3. The choice of the cell line is not appropriate for the study. The point of guided bone regeneration is to avoid any soft tissue and fibroblast, but the NIH-3T3 cell line was used. It is highly recommended to use a cell line with osteogenic lineage.
  4. An in vitro confocal or microscope images should be included to validate the cell attachment properties on the film.
  5. The discussion section should be written carefully as most of the text has little relevance in explaining the results.

Author Response

1. The primary focus of this study seems to improve the mechanical strength of gelatin by adding gellan gum. However, there was no experiment to show that the proposed approach increased the mechanical strength of the film. It is recommended to include data that shows the mechanical properties of the film before and after modification.
- The primary focus of this study was to evaluate the possible applicability of the film for guided bone regeneration. Like your valuable comment, it would have been good to meassure the mechanical properties of the film, but we did not target on mechanical properties.

2. The proposed film needed to be characterised thoroughly. In the manuscript, however, there was no data or explanation related to characterisation. It would be good to include the degradation studies in the revised manuscript.
- The degradation was observed in histological examination. Although the degradation rate may vary (Ilhan Yu et al./Procedia CIRP, 65 ( 2017 ) 78–83 ), we thought that degradation of gellan gum in body fluid is obvious.

3. The choice of the cell line is not appropriate for the study. The point of guided bone regeneration is to avoid any soft tissue and fibroblast, but the NIH-3T3 cell line was used. It is highly recommended to use a cell line with osteogenic lineage.
- We used the NIH3T3 cell to check cell viability. Since we believe that the integration of the membrane to overlying soft tissue is also important for stabilization of bone material, we used the NIH3T3 cells. But we also agree to your comment.

4. An in vitro confocal or microscope images should be included to validate the cell attachment properties on the film.
- Unfortunately, we did not have much time for study of cell attachment properties.

5. The discussion section should be written carefully as most of the text has little relevance in explaining the results.
- We modified the discussion.

Reviewer 3 Report

It would have been good if author could showed some viability and proliferation data with different time point to correlate with histology result.

There are some english errors correct them.

if possible add some references of Materials journal (ISSN 1996-1944)

Author Response

It would have been good if author could showed some viability and proliferation data with different time point to correlate with histology result.
-As your recommendation, it would have been good if there was comparison of the quality of newly formed bone. But our interest was focused on the effect of the GEL/TSG film as barrier to the soft tissue invasion into bone defect.

There are some english errors correct them.
-English erros are corrected.

if possible add some references of Materials journal (ISSN 1996-1944)
-We added reference of Material journal.

Reviewer 4 Report

The current manuscript entitled “Effect of gellan gum/tuna skin film in guided bone 2 regeneration in artificial bone defect in rabbit calvaria” is NOT designed and performed well. Authors tried to develop gellan gum with the combination of gelatin production from tuna skin which can be potentially applied for bone tissue repair and generation in the form of the protective membrane in bone-related surgery in guided bone tissue regeneration.

There are no significant results in the manuscript.

The author may rewrite the first sentence of abstract.

The abstract is too short and it can be rewritten with the essence of results and findings.

The literature of review is missing in the Introduction section.

Page no. 2 and line 49, author can be rewritten the sentence.

The conclusion is missing.

Overall, the current manuscript is not acceptable form in Materials Journal.

Author Response

There are no significant results in the manuscript.
-We prepared the mixture of gellan gum and tuna skin gelatin and fabricated in form of film to use as membrane for guided bone regeneration. We performed a MTT assay to identity cell viability of the film. It was non-toxic, and the survival rate of cells was 80% or more. And we performed animal study of rabbit model to investigate the feasibility to use the gellan gum/tuna skin gelatin film as GBR membrane. The grafted bone in the group treated with the GEL/TS membrane was well maintained in micro CT analysis and the bone regeneration was confirmed in entire experimental period in histologic examination. Please reconsider our manuscript for publication.

The author may rewrite the first sentence of abstract.
-We rewrote the first sentence of abstract.

The abstract is too short and it can be rewritten with the essence of results and findings.
-We rewrote the abstract with more of result and findings.

The literature of review is missing in the Introduction section.
-We added literatural review in introduction.

Page no. 2 and line 49, author can be rewritten the sentence.
-We rewrote the sentence.

The conclusion is missing.
- We separated the conclusion from discussion.

Round 2

Reviewer 1 Report

Dearest Authors,

thanks for having resubmitted your work in an updated version.

I still have some major concerns:

1) the lack of explicit references to possible traslability into human of these materials makes this work poorly interesting. You stated "Yet the result of GEL/TSG was fair, its suitability to human is planned in future after comparing with other materials". this should be widely cleared.

2) the lack of reference to normative framework and guidelines should be addressed and discussed in detail. 

3) still literature is not completely updated in terms of recent avances in this field.

Best

g

Author Response

1) the lack of explicit references to possible traslability into human of these materials makes this work poorly interesting. You stated "Yet the result of GEL/TSG was fair, its suitability to human is planned in future after comparing with other materials". this should be widely cleared.
- We tried to find references, but there was no definite reference that state the translability into human. Biocompatibility of GEL/TSG could be topic of future study.

2) the lack of reference to normative framework and guidelines should be addressed and discussed in detail.
- Development of biomaterial includes scientific study about biostability and clinical study about biocompatibility. The normative framework and guidelines for experiments seem to be necessary to check biocompatibility before application to human. There is many of natural polymers, which are considered as candidate of biomaterial for bone graft including GEL and TSG. Evaluating biocompatibilty of GEL/TSG could be the future study.

3) still literature is not completely updated in terms of recent avances in this field.
- We updated more references in discussion.

Reviewer 2 Report

  1. If the article is focused on 'the possible applicability of the film for guided bone regeneration.' implies that gellan gum/tuna skin film is a first-of-its-kind material for GBR application. But I don't feel it is reflected well in the manuscript text. what is novel in the study? 
  2. The cell attachment studies are a crucial step in the development of any new scaffold material. I don't understand the author's response 'we did not have much time for the study of cell attachment properties'. 

Author Response

1. If the article is focused on 'the possible applicability of the film for guided bone regeneration.' implies that gellan gum/tuna skin film is a first-of-its-kind material for GBR application. But I don't feel it is reflected well in the manuscript text. what is novel in the study?
- We developed GEL/TSG film for GBR appilcation. There is reference that used GEL for application (Chang et al./J Med Biol Eng, 30 (2010) 99-103). We believe that adding gelatin to gellan gum and using it as barrier for GBR could be the novel point. There is studies that used gellan gum and gelatin as scaffold. Scaffold for tissue engineering is to replace body structure. We used the GEL/TSG as barrier for GBR which is to be disappeared after bone regeneration.

2. The cell attachment studies are a crucial step in the development of any new scaffold material. I don't understand the author's response 'we did not have much time for the study of cell attachment properties'.
- As mentioned above, we used the GEL/TSG as barrier for GBR which is to be disappeared after bone regeneration. We agree to your comment emphasizing the importance of the cell attachement study, but we don`t think that is essential for the development of membrane or film for GBR. Inhibiting the growth of fibroblasts at the bone defect could be helful in GBR.

Round 3

Reviewer 1 Report

Dearest Authors,

I still believe you should at least discuss our work in the perspective of the mandatory biocompatibility requirements (see e.g.ISO10993 in its various sub chapters) and its relevance toward possible future clinical studies.

best

Author Response

I still believe you should at least discuss our work in the perspective of the mandatory biocompatibility requirements (see e.g.ISO10993 in its various sub chapters) and its relevance toward possible future clinical studies.
- Thank you for your valuable comment. We added our work in the discussion.

Reviewer 2 Report

I recommend the authors to include text in the introduction and cite those articles which reported gellan gum and gelatin as a scaffold.

Author Response

I recommend the authors to include text in the introduction and cite those articles which reported gellan gum and gelatin as a scaffold.
- Thank you for your valuable comment. We added that in the introduction.